# Impact of Enniatin and Deoxynivalenol Co-Occurrence on Plant, Microbial, Insect, Animal and Human Systems: Current Knowledge and Future Perspectives

**DOI:** 10.3390/toxins15040271

**Published:** 2023-04-06

**Authors:** Irene Valenti, Francesco Tini, Milos Sevarika, Alessandro Agazzi, Giovanni Beccari, Ilaria Bellezza, Luisa Ederli, Silvia Grottelli, Matias Pasquali, Roberto Romani, Marco Saracchi, Lorenzo Covarelli

**Affiliations:** 1Department of Food, Environmental and Nutritional Sciences, University of Milan, 20133 Milan, Italy; irene.valenti@unimi.it (I.V.); matias.pasquali@unimi.it (M.P.); marco.saracchi@unimi.it (M.S.); 2Department of Agricultural, Food and Environmental Sciences, University of Perugia, 06121 Perugia, Italy; milos.sevarika@unipg.it (M.S.); giovanni.beccari@unipg.it (G.B.); luisa.ederli@unipg.it (L.E.); roberto.romani@unipg.it (R.R.); lorenzo.covarelli@unipg.it (L.C.); 3Department of Veterinary Medicine and Animal Sciences, University of Milan, 26900 Lodi, Italy; alessandro.agazzi@unimi.it; 4Department of Medicine and Surgery, University of Perugia, 06132 Perugia, Italy; ilaria.bellezza@unipg.it (I.B.); silvia.grottelli@unipg.it (S.G.)

**Keywords:** mycotoxins, biological systems, co-exposure, synergism, antagonism, toxicity, *Fusarium*

## Abstract

*Fusarium* mycotoxins commonly contaminate agricultural products resulting in a serious threat to both animal and human health. The co-occurrence of different mycotoxins in the same cereal field is very common, so the risks as well as the functional and ecological effects of mycotoxins cannot always be predicted by focusing only on the effect of the single contaminants. Enniatins (ENNs) are among the most frequently detected emerging mycotoxins, while deoxynivalenol (DON) is probably the most common contaminant of cereal grains worldwide. The purpose of this review is to provide an overview of the simultaneous exposure to these mycotoxins, with emphasis on the combined effects in multiple organisms. Our literature analysis shows that just a few studies on ENN–DON toxicity are available, suggesting the complexity of mycotoxin interactions, which include synergistic, antagonistic, and additive effects. Both ENNs and DON modulate drug efflux transporters, therefore this specific ability deserves to be explored to better understand their complex biological role. Additionally, future studies should investigate the interaction mechanisms of mycotoxin co-occurrence on different model organisms, using concentrations closer to real exposures.

## 1. Introduction

*Fusarium* head blight (FHB) is one of the most widespread and damaging fungal diseases of common and durum wheat, as well as other small-grain cereals [1], caused by species of the genus *Fusarium* [2]. It is able to impair grain yield and quality due to mycotoxin accumulation. *Fusarium* species distribution is usually related to agricultural practices, cultivar susceptibility, climatic conditions (especially at wheat anthesis), and fungicide application [3,4,5,6,7,8]. For this reason, the composition of the species involved in the FHB complex is dynamic [9]. Generally, *Fusarium graminearum* is considered the most important and aggressive FHB causal agent [10]. However, other species such as *Fusarium culmorum*, *Fusarium avenaceum,* and *Fusarium poae* are very often detected in many cultivation areas across the world [11,12,13,14]. *Fusarium* species associated with FHB can biosynthesize a wide range of mycotoxins and secondary metabolites with toxic effects on animals and humans [15]. Among them, trichothecenes are subject to extensive studies due to their toxicity and frequent occurrence [16]. They are sesquiterpenoid mycotoxins and are divided into A and B groups, characterized by different hydroxyl groups in the C-8 position of the trichothecene backbone [17]. Deoxynivalenol (DON) is chemically known as (3α,7α)3,7,15-trihydroxy-12,13-epoxytrichothec-9-en-8-one. It is a cyclic sesquiterpenoids epoxide that contains three hydroxyl groups at C-3, C-7, and C-15 and a carbonyl function at the C-8 of the 12,13-epoxytrichothec-9-ene core [18]. DON with its acetylated derivatives (3-acetyl deoxynivalenol, and 15-acetyl deoxynivalenol), is principally produced by *F. graminearum* and *F. culmorum* and is considered the most common trichothecene detected in cereals worldwide [19,20,21].

The International Agency for Research on Cancer (IARC) has classified DON in Group 3, so it is not classifiable as carcinogenic to humans [22]. However, the ingestion of DON in mammals can result in acute toxic effects such as nausea, gastroenteritis, vomiting, diarrhea, and increased salivation. In addition, chronic toxic effects such as immunotoxicity, altered nutritional effects, weight loss, and anorexia have been frequently observed. However, these effects of DON ingestion may differ depending on the metabolism, absorption, and elimination mechanisms of different organisms [20,23,24,25]. Therefore, the European Union (EU) has set maximum levels for several mycotoxins, including DON, in various food matrices, such as raw cereals and some derived products for human consumption [26]. In addition, also other countries, such as China, Russia, Brazil, the USA, Canada, and Japan, have also indicated or are indicating DON tolerable limits in several raw cereals and derivatives [16,27,28].

In accordance with the data collected, a very high incidence of samples positive for the presence of DON in wheat has been observed worldwide [29]. In some cases, very high values have been detected in wheat grains coming from different countries. In addition to these extreme values, samples were also shown to be often above the legal limit in baby food, pasta, and noodles [20,29,30]. High DON contamination has been detected not only in wheat but also in other cereals, such as barley, oats, and maize samples [15,16,29,30].

In addition to DON, several data published in the last decades have shown an increasing incidence of other *Fusarium* secondary metabolites, also known as emerging mycotoxins [31]. Among them, enniatins (ENNs) are very common worldwide in wheat, barley, and other cereals, and their derivatives for human and animal consumption [32,33,34,35]. ENNs are N-methylated cyclic hexadepsipeptides composed of alternating residues of N-methyl branched-chain amino acids, and hydroxy acids [36]. Due to the pore-like structure of the cyclodepsipeptide ring of ENNs, they possess ionophoric properties. Electrophysiological analyses showed that they can be easily incorporated into the cell membrane and form passive cation-selective channels evoking changes in intracellular ion concentration. This property may explain the broad range of biological activities attributed to ENNs [37].

To date, at least 29 different analogs have been characterized, but only a few of them are generally detected in cereals: enniatin A (ENA), enniatin A1 (ENA1), enniatin B (ENB) and enniatin B1 (ENB1) [32]. In turn, within these four analogs, ENB and ENB1 showed the highest levels in cereal grains in many cultivation areas both in terms of concentration and occurrence [11,38,39,40,41,42,43]. ENNs are mainly produced by members of the *Fusarium tricinctum* species complex (FTSC), such as *F. avenaceum* and *F. tricinctum*. Despite their frequent occurrence worldwide, to date, ENNs have not yet been included in any regulation because their proprieties and impact on humans and animals are still unclear [31,44]. In 2014, a scientific opinion from the European Food Safety Authority (EFSA) on the risks to human and animal health related to ENN presence in feed and food was published. However, given the lack of toxicity data, no conclusions on toxic exposure were drawn [45]. Nevertheless, since a concern due to possible interactions with other mycotoxins and chronic exposure was highlighted [45], regulation could be evaluated in the next future.

The single-field coexistence of different *Fusarium* species is very common [9,46,47,48,49,50] and, consequently, a wide range of *Fusarium* mycotoxins can be present within a single-grain sample collected from the same field. Due to the high worldwide diffusion of *F. graminearum* and *F. avenaceum*, according to data collected in many surveys, co-occurrence of ENNs and DON is common in raw samples, food, or feed [11,48,51,52,53,54,55,56,57,58,59,60].

While DON possesses a well-studied activity towards plants [61], insects [62,63], animals, and humans [19], ENNs started to attract researchers’ attention in the last few years. Some studies, for example, have begun to elucidate their role in fungal virulence [64], in the *in vitro* interaction with other FHB causal agents [65], and their impact on animals and humans [66]. However, little is still known about ENN’s role in different systems and, in particular, about their interactions with major mycotoxins such as DON.

For this reason, considering the frequent ENN and DON co-occurrence, this paper aims to review the information already published that can be useful in understanding the combined effect of the two mycotoxins. Specifically, the effects of ENN and DON combination were described on: fungal virulence towards the host; competition among FHB causal agents; wheat microbiota; insects; dairy cows; humans. For each system mentioned, missing aspects and what could be conducted to better clarify the combined role of ENNs and DON is outlined.

## 2. Effects of ENN and DON Co-Occurrence on Biological Systems

### 2.1. Host Plants

The mycotoxin DON is well-known both for its role as a virulence factor [67], and for its phytotoxic activity. In various plant species, DON is a potent protein synthesis and cell division inhibitor and causes a significant mitosis reduction, especially in wheat and bean [68,69]. DON strongly inhibits coleoptile and shoot elongation in wheat [70], and also negatively affects root growth [65,71,72]. Contradictory results regarding DON activity on cell death are available in the literature. For example, on wheat was shown that treatments with variable concentrations of this mycotoxin induced oxidative stress, accumulation of hydrogen peroxide, and apoptosis-like programmed cell death (PCD) [61]. It was reported that exposure of Arabidopsis leaves to DON caused the inhibition of plant antioxidant systems, resulting in an oxidative burst and an increase in lipid peroxidation [71]. Instead, other studies showed suppression of PCD by DON, mainly at low concentrations [65,73]. Treatments with DON in wheat genotypes caused alterations in carbohydrate and protein metabolism. This resulted in increased free amino acids, probably derived from irregular protein hydrolysis or related to an active plant response induced by the same mycotoxin [74]. On the other hand, a potential role as a defense priming molecule has been documented for DON or its masked forms [75,76].

In contrast to DON, little is known about the effects of ENNs on plants. Previous studies reported the inhibition of germination and the induction of plant wilting caused by these mycotoxins [77,78]. More recent studies showed that ENB affected the virulence of *F. avenaceum* in potato tubers but not in durum wheat and pea [64]. The only study conducted *in planta* about the effects of DON and ENB co-occurrence demonstrated their synergistic activity in inhibiting germination, growth, and chlorophyll degradation. Conversely, they acted antagonistically relative to cell death, which was significantly induced by ENB and counteracted by DON [65]. Furthermore, a pilot study reports that treatments with ENB reduced the antioxidant capacity in wheat, confirming the role of this mycotoxin in the induction of oxidative stress [79].

The presence of different mycotoxins in cereal grains is currently increasing [51,52,53,80,81]. For this reason, investigations regarding the effects of DON, ENB, and their association in plant tissues, and in the virulence of *F. graminearum* and *F. avenaceum* would be desirable. In addition, elucidating the mode of action of DON and ENNs in defense priming may be an important advancement for future understanding and enhancement of the immune response to diseases of important plant species such as wheat.

### 2.2. Fusarium Head Blight Causal Agents

Many plant species can often be simultaneously infected by more than one pathogenic species [82]. For this reason, the impact of plant diseases is generally not the result of a single species/single strain infection but the consequence of a multispecies interaction of more pathogens. They may coexist, taking advantage or competing with each other in a specific biological niche [83,84]. In these interactions, fungal metabolites may protect producing fungi against other microorganisms and help in realizing a more suitable environmental niche [85,86]. The coexistence of many FHB species is common in grain coming from one field, with wide variability among species [5,49,87,88,89]. However, different *Fusarium* species can co-exist also in the same niche, such as the wheat head [90,91,92]. The co-occurrence of more *Fusarium* species in the same head means a significant increase [93] or decrease [91,94] in mycotoxin contamination.

Despite these fluctuations, secondary metabolites may play a crucial role in the possible synergistic or antagonistic relationships among *Fusarium* species within the same plant tissue (head).

Due to their wide diffusion and co-occurrence at the field level [48,87,95], *F. graminearum* and *F. avenaceum* may coexist in the same head. Their main mycotoxins, DON and ENNs, respectively, could regulate the interactions between these two pathogens with other *Fusarium* species.

Generally, DON biosynthesis by *F. graminearum* may facilitate the pathogen during competition with other eukaryotic organisms [67]. However, few studies explore the toxicity of DON on the microbiota [96]. Recently, it has been observed that DON promotes *F. avenaceum* growth *in vitro* [65] showing that it could not be an important factor in *Fusarium* competition, but only a strategic compound in disease development in wheat [61]. In addition, other authors [97,98] suggested a negligible role of DON in fungal interactions with non-*Fusarium* fungi.

ENNs have always been considered compounds acting as enzyme inhibitors and immunomodulators [99]. These compounds showed antimicrobial activity against some fungi [100], and bacteria [101]. Nevertheless, no evidence of *Fusarium* growth inhibition was observed as a direct effect of ENB [100]. However, recently, a negative interference of ENB on *F. graminearum in vitro* development was observed, and an advantage on *F. avenaceum* growth was also reported. Conversely to DON, ENNs seem not to be fundamental for FHB progression in wheat [64], but they could have an important role in interspecific competition [65]. A synergistic effect was observed with the co-presence of DON and ENB in reducing *F. avenaceum* and *F. graminearum* growth [65].

Given the high frequency of ENN and DON co-occurrence, future studies should focus on the role of this combination in *F. avenaceum* and *F. graminearum* competition with other species composing the FHB community. In detail, the *in vitro* activity of ENNs+DON towards the main FHB species (*F. graminearum*, *F. culmorum*, *F. poae,* and *F. avenaceum*) could be investigated by evaluating the possible fungal growth inhibition/stimulation. In addition, to determine a possible ENN+DON activity on the synergism/competition between *Fusarium* species *in planta*, head co-inoculation could be performed. Ideally, *F. avenaceum* and *F. graminearum* mutants unable to produce ENNs and DON, respectively, should be used.

### 2.3. Microbiota

Natural microorganisms colonizing a specific environment such as a cereal field, display a key role in the plant’s growth [102]. The microbial heterogeneity includes archaea, bacteria, cyanobacteria, fungi, and protozoa [103]. Some of these soil and plant-associated microbes bring beneficial advantages to the plants by improving their fitness and productivity [104]. Several microbiota members can be used as biological control agents (BCAs) for their active competition limiting pathogens’ growth and their ability to produce unsafe secondary metabolites such as mycotoxins [105]. For example, several studies have described promising results in reducing the FHB incidence and DON production by the bacterial genera *Streptomyces* [106,107,108], *Bacillus* [109,110], *Cryptococcus* [111], and *Pseudomonas* [112]. Bacteria can also induce mycotoxin detoxification by biosorption or biodegradation [113,114]. DON can be reduced *in vitro* from 43% up to 86% by the microbial flora coming from animal stables and wheat fields [115]. Instead, the ENNs can be degraded by probiotic bacterial strains up to 99% [116]. Although some microorganisms have been described as mycotoxin degraders *in vitro* [117], mycotoxin biodegradation is still an interesting challenge.

To date, the well-known antibacterial property of ENNs was tested against a wide range of both Gram-positive and Gram-negative human pathogens reporting an IC_50_ > 10 µg/mL [118]. This level was significantly higher than the recently detected environmental concentrations [81]. ENNs were also described to have antimicrobial properties against *Mycobacterium tuberculosis* [119], *Plasmodium falciparum* [120], *Candida albicans* [121], and other human pathogens [122]. Interestingly, in *Saccharomyces cerevisiae* ENNs showed an inhibitory capacity towards transmembrane *Pdr5p* pump (involved in the multidrug resistance mechanism) [123], suggesting their potential effect in modulating xenobiotic efflux. The antagonistic effect of ENB was also investigated on some fungal species such as the BCAs *Trichoderma harzianum* and *Beauveria bassiana*, showing a minimum inhibitory concentration (MIC) value of 1 and 5 µg, respectively [100]. Conversely, the fungal pathogens belonging to the genera *Fusarium*, *Aspergillus,* and *Penicillium* showed no sensitivity to the highest concentrations tested.

To date, most of the analyses performed on ENNs have considered just acute toxicity. Focusing on the data about the microorganisms’ sensitivity (Appendix A), more than 40% of organism models showed no effect at the highest concentrations used, greater than those reported in natural contaminations [46,81,95,124,125,126]. According to these data, microorganisms showed dissimilar sensitivity from 2000 µg to 10 ng or in the range of 75–0.2 µg/mL. Moreover, the most sensitive microorganism was *Plasmodium falciparum* K1, showing an IC_50_ from 1.9 to 0.2 µg/mL depending on the type of ENN [120]. In this regard, the ENN category is another variable in the results. For example, *Staphylococcus aureus* CECT 240 showed no sensitivity when it was exposed to 2000 µg of ENB [122]. On the other hand, a MIC value of 1000 ng and 10 ng was detected considering ENJ1 and ENJ3, respectively [127]. Moreover, *Bifidobacterium adolescentis* 5871 exhibited toxicity effects just to ENB1 and not to ENA, A1, and A2 [101]. Most of the data reported in Appendix A are focused on human pathogens and probiotic bacteria, excluding BCAs or competing pathogens, considered key role organisms in preventing FHB disease. Additionally, DON’s impact on microorganisms has been poorly investigated. The antibiofilm activity of DON was detected in *C. albicans*, but not in *C. tropicalis*, *E. coli*, *A. tumefaceiens*, *S. aureus,* and *P. aeruginosa* [128]. In addition, a recent study has shown no significant effects *in vitro* on *Bacillus* strains grown in the presence of different DON concentrations [96]. Interestingly, some studies suggest a DON influence on both intestinal pig microbiota [129] and soil microflora altering the community structure [130].

Despite all these data, the effect of both mycotoxins on wheat microbiota should be further investigated. The impact of DON and ENB, ENB1, and beauvericin mixture (EB), alone or in combination, was examined on piglets’ gut microbiota where a microbial pattern alteration was observed in all treatments. However, only EB led to a significant decrease in microbiota diversity [80]. On the other hand, the ENN–DON co-occurrence on wheat microbiota is still unexplored.

Therefore, given the important role of the microbial community in plant vitality, a better understanding of the interactions between microbiota and mycotoxins may contribute to more sustainable crop management practices. In particular, the effect of ENNs and DON on the overall endophytic and surface colonizing populations of wheat, and the role of these mycotoxins during the interaction of pathogens with BCAs could be further investigated. This could allow deciphering how molecular interactions can shape the plant microbiome.

### 2.4. Insects

*Fusarium* species are important fungal pathogens that infect plants and are also exploited by various insects. Thus, insects and pathogens are often exposed to each other and can directly or indirectly interact and therefore affect the same host plant. Currently, there is no information about a possible mechanism of biotoxic action. However, insects are able to metabolize and degrade mycotoxins ingested during the developmental stage by exploiting different enzymatic detoxification mechanisms (mainly based on Cyt P450-enzyme and NADPH) [131,132].

Most of the studies carried out, showed a correlation between insect activity and the level of *Fusarium* infection and/or mycotoxin accumulation. Wheat plants infested with aphids and infected with *F. graminearum* showed significantly more symptoms after six days of inoculation [133]. Moreover, the plants infested with the aphid *Sitobion avenae* and influenced with *F. graminearum* expressed a two- and fivefold increase in the amount of pathogen DNA and DON, respectively [63]. Recent findings showed that the timing of aphid colonization has a lower effect on disease severity [63]. Similar results were reported in the case of *F. graminearum* and lepidopteran injury in maize, where insect damage to cobs resulted in elevated DON accumulation and disease incidence [134,135]. The wounds made in late ear development and to the side of the ear had higher effects than those during silking, kernel establishment, silk clipping, tip injury, or kernel grazing [134]. In the case of aphids feeding on wheat plants infected with *F. graminearum*, higher aphid mortality was reported due to higher DON concentrations; thus, aphids tend to reside and develop on plant heads devoid of fungal infection [63,133]. *Fusarium* species interact with plants by changing their chemical volatile profile. Plants infected with *Fusarium* species were repellent towards aphids when tested in a Y-tube olfactometer, due to the 2-pentadecanone compound [63], resulting from the presence of *Fusarium* mycotoxins [62]. In addition, aphids also interact with plants by inducing defense genes, thus provoking earlier and enhanced sensitive responses of plants against *Fusarium* species [133].

The complex insects–fungi/mycotoxins interactions have been shown by the DON effect on the parasitic wasp of the aphid *S. avenae*. In this study [136], the sublethal and lethal effect of DON on *S. avenae*, and the subsequent effect on its parasitoid *Aphidius ervi*, in terms of decreased offspring production, were demonstrated.

Direct effects of mycotoxins on insects were mainly observed on insects used for food and feed. The species used as model organisms were Dipterans (*Hermetia illucens*) and Coleopterans (*Alphitobius diaperinus* and *Tenebrio molitor*). The most studied mycotoxin was DON, whose effect differed from species to species. In the case of *H. illucens* and *Spodoptera frugiperda*, no effect on its larval biomass was recorded, while *T. molitor* and *H. zea* exhibited significantly lower growth performance when exposed to DON [137,138,139]. DON had a significant effect on the mortality of the aphid *Acyrthosiphon pisum*, while no or slight effect (<10%) was observed in other insect species (*H. illucens*, *T. molitor*, *S. avenae*) [138,140,141,142,143,144]. Moreover, these insects showed low levels of mycotoxin accumulation. In the case of *H. illucens* no mycotoxins could be detected when fed with a diet containing high levels of DON (up to 125,000 µg/kg) [143,145]; a similar effect was recorded also for *T. molitor* larvae when exposed to 12,000 µg/kg [141,142,144].

On the other hand, only a few studies investigated ENN toxicity in insects (Appendix A) after ENN purification from *Fusarium lateritium* cultures. For example, ENN insecticidal properties on the lepidopteran *Choristoneura fumiferana* [146,147] were observed, contrary to what was observed in the case of *Galleria mellonella* [148]. Moreover, mycelial extracts from *Cordyceps fumosorosea*, containing also ENNs, expressed insecticidal activity towards *Bemisia tabaci* and *Aphis craccivora* [149]. In addition, *T. molitor* fed on wheat kernels colonized with *F. avenaceum* and *F. culmorum* showed a significantly higher mortality rate (even though not substantial) [150].

To date, no information on the effect of the combination of the two mycotoxins on insects is available. For this reason, specific data on the role played by DON and ENNs on entomophagous insects referring to the wheat–*Fusarium*–aphid biological system would be desirable.

### 2.5. Dairy Cows

Multiple mycotoxins can occur in both forages and concentrates for animal nutrition, so the relative carry-over in animal-derived products represents a huge concern for animal health and food safety [151].

The presence of ENNs in livestock feedstuff [152], cereals [34,41,43,66,153,154,155] and by-products [156,157] has been extensively outlined during past years. As an example, some researchers reported as more than 78% of the analyzed maize silage samples were contaminated by ENNs [40], with the most abundant ones represented by ENB and ENB1. The co-occurrence of ENNs with DON was often reported at ranges included between 58% and 61% of the analyzed samples [57]. A recent survey showed how increased content of emerging mycotoxins could be accompanied by high DON content in mixed infections [154] and, in some cases, the presence of DON and ENNs at the same time reached 100% of the analyzed samples [52].

To date, few conclusions for the *in vivo* effects of multiple mycotoxin contamination are available. However, undesirable effects in ruminants are often related to low feed intake and rumination activity, immunosuppression, and increased pro-inflammatory cytokines [158], leading to subclinical and not specific health problems and impaired milk production [159]. Negative effects are more pronounced in high-yielding dairy cows fed with high fermentable diets [160], because of microbial shifts in the rumen [161] and consequent impairment of mycotoxin detoxification by the resident microbiota [162]. The co-occurrence of ENNs with DON is reported in feeds [60,163] and, consequently, possible synergistic, additive, or antagonistic effects on animals can be hypothesized. However, so far, only *in vitro* studies have been conducted [53,164,165]. Given the lack of *in vivo* trials on ENN and DON co-occurrence, we can only speculate on the possible effects resulting from the simultaneous presence of these two mycotoxins. DON in ruminants leads to gastrointestinal disorders, and immunosuppression, with decreased feed consumption and lower performance [25,166,167]. These effects were due to a shift of energy metabolism available for production to sustain immune system depression and increased inflammation [159], together with an induced ruminal dysbiosis and increased permeability of the rumen and/or gut epithelia [168]. A study [53] showed that ENN and DON co-occurrence did not change the toxicity of DON itself. In addition, limits for ENN concentration in the diets of ruminants have not been established. However, a recent *in vitro* study showed that over 70% of ENB was degraded after 48 h under ruminal physiological pH [162]. On the other hand, the same authors reported that, in the case of a subacute rumen acidosis, ENB degradation was inhibited, outlining how a portion may pass to the intestine under altered rumen conditions. The carry-over of ENNs into milk may be possible but, to date, it has only been detected at very low levels in sheep milk [169].

However, no data on the occurrence of these emerging mycotoxins in bovine milk are currently available.

### 2.6. Humans

*Fusarium* mycotoxins contaminate several products destined for human consumption. Consequently, they can be absorbed through the gastrointestinal tract resulting in biological effects on different tissues. According to different studies, contaminated cereal foods, including baby food and gluten-free pasta, contained at least one mycotoxin. ENN–DON co-occurrence has often been reported, even if with dissimilar proportions with levels ranging from 0.03 to 710 μg/kg for ENB and from 16 to 295 μg/kg for DON [170,171,172,173].

Mycotoxin contamination also regards non-cereal-based food such as milk thistle (ENB up to 8340 μg/kg, DON up to 5958 μg/kg) [174], tea (ENB up to 9260 μg/kg, DON up to 2890 μg/kg) [175,176]. In addition, eggs and meat can be contaminated by mycotoxins (ENB up to 15 μg/kg, DON up to 0.79 μg/kg), suggesting that, although marginally, animal-derived foods can contribute to human mycotoxin exposure [177,178].

The resistance of mycotoxins to food processes has been reported, although with contradictory results. For example, a reduction of up to 80% in drying pasta cooked at 70–90 °C was detected [179]. On the other hand, 60% of DON and 83–100% of ENNs were retained in samples of cooked pasta [173].

More importantly, upon ingestion, mycotoxins can be found in tissues and in body fluids. The wastewater-based epidemiology is a biomonitoring approach that provides direct information on human exposure to food contaminants. The analyses of 29 samples collected in Latvia revealed that ENB can be detected in more than 86% of samples and DON was found in all the samples [180]. The analyses of mycotoxin presence in 24 h urine samples and serum of both vegans (*n* = 36) and omnivores (*n* = 36) revealed that ENB in serum and DON glucuronide in urine were detected in 57–90% of samples, with no significant differences between diets. The presence of mycotoxins in the blood and urine of 3000 Swedish adolescents revealed that 4.8% of urine samples were positive for DON and 99.2% of blood samples contained ENB [181]. Both DON and ENB were also detected in breast milk [182,183,184] and in infants’ urine suggesting gut absorption [185]. All these data strongly support the hypothesis of mycotoxin bioaccumulation in tissues which might result in chronic low-dose exposure. For this reason, the ENN–DON co-occurrence in foods and body fluids makes the understanding of their combined effects of great importance for human health.

Several reports examined the *in vitro* effects of single mycotoxin exposure on human cells, but only a very limited number of them investigated the combined effects of ENN and DON on experimental models based on human cell lines.

The combined effects of ENB and DON on cell proliferation were explored in the colorectal carcinoma cell line Caco-2 after 24 h of incubation. The IC_50_ values were 6.3 μM and 13.0 μM for ENB and DON, respectively. When the mycotoxins were used in a 1:2 proportion an antagonistic response was detected [186]. A 72 h incubation of Caco-2 cells with ENB or DON led to a significant reduction in cell viability with an IC_50_ of 3.9 μM and 5.5 μM, respectively. The cell viability decreased significantly during the co-exposure in a 1:1 proportion. The results showed synergism when mycotoxins were used at IC_75_ or IC_90_ concentrations for 48 h [164].

The cytotoxic effects of a 24 h ENB and DON exposure was examined on SH-SY5Y human neuroblastoma cells, resulting in a calculated IC_50_ of 0.43 μM and 0.94 μM, respectively. Moreover, the co-exposure in a 1:5 ratio resulted in a cytotoxic effect superimposable to that produced by DON alone. In the presence of DON, antagonism was observed also in this cell model [187].

The antagonistic response of DON could be due to its ability in enhancing Aryl hydrocarbon receptor (AhR) expression and activation [188]. In fact, AhR mediates the upregulation of xenobiotic metabolizing enzymes and drug efflux transporters, including ABC transporters [189], involved in the export of ENB out of the cell to mitigate its cytotoxicity [190]. Thus, we may consider that the activation of AhR by DON might result in its antagonistic behavior against ENNs.

Given the frequent co-contamination of foods with ENNs and DON, further studies are urgently needed to better define the effects of chronic mycotoxin co-exposure. In particular, the analyses should examine the integrity of the intestinal epithelial barrier, the hepatic metabolism, and the immune response in order to obtain a better assessment of the risk to human health.

## 3. Discussion

Mycotoxins are among the major threats to food safety and consequently to the health of related biological systems. The presence of ENNs in the world [11,33,81,125,191] and the co-occurrence of different mycotoxins in cereal grains, are currently increasing [51,52,53,80,81].

This work illustrates that, in contrast to DON, little is known about the impact of ENNs and even less about their co-occurrence on biological systems. This review provides an overview of this double exposure, considering that DON is the most frequent mycotoxin in cereal crops [192,193,194], and ENNs, among the non-regulated mycotoxins, are present in many field surveys. Therefore, looking at the effects of the ENN–DON interactions would allow a better understanding of the complex biological effects of secondary metabolite combinations on different biological systems.

Usually, a toxicological evaluation is based on individual mycotoxin and a single model system. However, living organisms, humans included, continuously interact with each other and with the environment, and are exposed to a mixture of toxic or potentially toxic compounds. Hence, a community-level overview of multi-contaminations is required to outline a more correct investigation for an appropriate risk assessment. In this regard, Table 1 summarizes the data about the effects of ENN–DON co-occurrence. Very few studies analyzed the consequences of mycotoxin co-exposure and most of the studies shown in Table 1 are focused on animal cell lines. Thus, it would be important to improve the knowledge of the key role organisms that directly or indirectly are affected by these mycotoxins. Data from the literature show the significance of analyzing the effects of combined mycotoxins which could show synergistic and/or antagonistic behaviors [53,65,164,186,187]. The ENB–DON co-exposure revealed a synergistic effect in both *F. graminearum* and *F. avenaceum* growth [65]. On the other hand, this co-occurrence showed both synergistic and antagonistic activity in *Triticum aestivum* [65] and in human colonic Caco-2 cells [164]. Moreover, data on both SH-SY5Y human neuroblastoma cells [187] and IPEC-1 porcine intestinal cell line [53] showed that the toxicity of, respectively, ENB–DON and ENB1–DON simultaneous exposure, was similar to the toxicity of DON alone.

ENN and DON’s ability in modulating drug efflux transporters is another important factor arising from the literature data [123,188,189] that shall be considered for fully assessing their biological role. Indeed, this activity could influence the uptake of some xenobiotics, improving the interactions and consequently the complexity of their effects.

Considering the literature reports analyzed, future studies should try to fill some knowledge gaps:(i)Understand the interaction mechanisms during mycotoxin co-occurrence;(ii)Increase data availability on the effects of ENNs and DON considering unexplored taxonomic or functional groups of organisms;(iii)Extend the dosage of mycotoxin concentrations tested to better simulate natural contaminations.

On the other hand, it should be considered that the interactions between mycotoxins and biotas or environmental matrices and other compounds could alter their chemistry and bioavailability, making the predictions more complex to model.

## 4. Conclusions

By exploring the literature data, we understand that several plant pathogens can be present at the same time in cereal fields. Consequentially, the grains may be contaminated by a mix of mycotoxins, mainly including ENNs and DON. Both mycotoxins can express their potential toxicity on multiple organisms. Their risk assessment is often carried out by exploring the effects of single contaminants. Indeed, the response to ENN–DON co-exposure is largely unexplored in key role organisms of the food chain such as insects, dairy cows, and plant microbiota (biocontrol agents included). Thus, further investigations would be required to complement the recent knowledge advancements on human and animal cells, wheat, and its fungal pathogens.

## Figures and Tables

**Table 1 toxins-15-00271-t001:** *In vitro* toxicity studies of ENN–DON co-occurrence.

Species/Cell Line	Mixture	Dose-Effect Parameters	Exposure Time	Interaction	References
*Fusarium avenaceum*	ENB + DON	100 mg/kg *	96 h	Synergism	[65]
*Fusarium graminearum*	ENB + DON	10 mg/kg *	96 h	Synergism	
*Triticum aestivum* A416 (seeds, seedlings)	ENB + DON	10 mg/kg + 10 mg/kg *	24 h	Both synergism and antagonism	
IPEC-1 intestine piglet cell line	ENB + DON	0.06 µM + 5.6 µM *	48 h	ND	[53]
		0.13 µM + 1.9 µM *			
		1.9 µM + 5.6 µM *			
	ENA1 + DON	0.03 µM + 5.6 µM *	48 h	ND	
		0.2 µM + 5.6 µM *			
		0.24 µM + 1.9 µM *			
	ENB1 + DON	0.06 µM + 5.6 µM *	48 h	ND	
		0.5 µM + 5.6 µM *			
		0.65 µM + 1.9 µM *			
Intestinal Caco-2 cells	ENB + DON (1:1)	5.59 µM	24 h	Antagonism at IC_10_ and IC_25_	[164]
				Additivity at IC_50_, IC_75_ and IC_90_	
		4.05 µM	48 h	Antagonism at IC_10_ and IC_25_	
				Additivity at IC_50_	
				Synergism at IC_75_ and IC_90_	
		4.33 µM	72 h	Antagonism at IC_10_, IC_25_, IC_50_, IC_75_ and IC_90_	
	ENB + DON	5 µM + 10 µM	24 h	Antagonism	[186]
Neuroblastoma SH-SY5Y cells	ENA + DON	0.15µM + 0.75 µM *	24 h	Antagonism	[187]
	ENB + DON	0.15 µM + 0.75 µM *	24 h	Antagonism	

* = MIC value; ND = not detected.

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
