# Peer review of "Impact of Enniatin and Deoxynivalenol Co-Occurrence on Plant, Microbial, Insect, Animal and Human Systems: Current Knowledge and Future Perspectives"

_toxins, 2023, doi:10.3390/toxins15040271_

Round 1

Reviewer 1 Report

The manuscript presents a summary of the toxicity of enniatin and deoxynivalenol co-occurrence based on published data, and revealed the need for further studies to elucidate the interaction/toxicity mechanisms of enniatin and deoxynivalenol co-occurrence on different model organisms. The authors should read through the manuscript and correct minor grammatical and typographical errors.

Comments

Line 25, delete “of ’’

Line 25, delete “quantity’’

Line 31, delete “also”

Line 51-53, correct sentence

Line 71, change “amount” to “concentration”

Line 127-130, 198-200, 283-285, correct sentence

Line 205, 293,427, correct error in spelling

Author Response

The manuscript presents a summary of the toxicity of enniatin and deoxynivalenol co-occurrence based on published data, and revealed the need for further studies to elucidate the interaction/toxicity mechanisms of enniatin and deoxynivalenol co-occurrence on different model organisms. The authors should read through the manuscript and correct minor grammatical and typographical errors.

The authors are very grateful to reviewer 1 for his/her revision that has surely allowed us to greatly increase the level of the manuscript.

  • Line 25: delete “of ’’

Done.

  • Line 25: delete “quantity’’

Done.

  • Line 31: delete “also”

Done.

  • Line 51-53: correct sentence

Corrected

  • Line 71, change “amount” to “concentration”

Done.

  • Line 127-130, 198-200, 283-285, correct sentence

Corrected

  • Line 205, 293,427, correct error in spelling

Done.

Reviewer 2 Report

The review highlights the necessity of research about Deoxynivalenol-Enniatin's synergic effect. A great number of references were evaluated and the sum of their information was good. It would be interesting to comment on their estructural properties that can be associated with the lack of studies of their simultaneous effects.

Author Response

The review highlights the necessity of research about Deoxynivalenol-Enniatin's synergic effect. A great number of references were evaluated and the sum of their information was good.

The authors are very grateful to reviewer 2 for his/her revision that has surely allowed us to greatly increase the scientific level of the manuscript.

It would be interesting to comment on their structural properties that can be associated with the lack of studies of their simultaneous effects.

Thanks for this comment. We added, in the introduction section, some information on structural properties of DON and ENNs as an integration to what was already present in the manuscript. However, we are not able to associate these properties with the lack of studies on their simultaneous effect.

Reviewer 3 Report

Impact of enniatin and deoxynivalenol co-occurrence on plant, microbial, insect, animal and human systems: current knowledge and future perspectives

The manuscript was reviewed for consideration in toxins. The topic is of interest with readers of food safety especially those whose research focus is of mycotoxins. The Contents are quite concise for example,

Section 2.5: Dairy cows, the detail of DON in cows fodder would be helpful to draw the intention of readers on the toxic presence of this toxin. A table would be very informative in this case.

Similarly, Humans. The incidents in human contamination of DON in the form of table would be ideal for a review like this

Conclusion

This part is so wordy only include the main highlights which would draw from this review article.  

Author Response

Impact of enniatin and deoxynivalenol co-occurrence on plant, microbial, insect, animal and human systems: current knowledge and future perspectives. The manuscript was reviewed for consideration in toxins. The topic is of interest with readers of food safety especially those whose research focus is of mycotoxins.

The authors are very grateful to reviewer 3 for his/her suggestions that greatly improve the manuscript quality.

The Contents are quite concise for example:

  • Section 2.5: Dairy cows, the detail of DON in cows fodder would be helpful to draw the intention of readers on the toxic presence of this toxin. A table would be very informative in this case.

Thanks for this comment. As reported in the title of the manuscript, we focused the attention of this paper on the co-occurrence of DON and ENNs. However, taking into account your request, we decided to include in section 2.5 some information about the presence of ENNs and their co-occurrence with DON considering what is currently available.

  • Similarly, Humans. The incidents in human contamination of DON in the form of table would be ideal for a review like this

Thanks for this comment. As reported for the previous comment, since the review focuses on DON-ENNs co-occurrence, we added in section 2.6 relevant information on the amount of DON and ENNs simultaneously found in foodstuff.

  • This part is so wordy only include the main highlights which would draw from this review article.

Also following the suggestion of reviewer 4, we added the section of Discussion in which we interpretated the main results and hypothesized future works. Now the Conclusions are shorter and include just the main highlights. Thank you for your suggestion.

Reviewer 4 Report

This interesting article deals with frequent Fusarium mycotoxins contaminating cereals all over the world - deoxynivalenol (DON), one of the major mycotoxins, and enniatins (ENNs), group of the emerging mycotoxins, providing an overview of the effects of their (simultaneous) exposure, emphasizing the need for more elaborate investigation of the interaction mechanisms of mycotoxin co-occurrence.

The article seems to be well prepared, with the introduction providing enough background on the subject. The reference list is substantial, not entirely current but relevant, while English language and style are generally satisfying. The Authors could consider to shorten the Conclusions section for more clarity, perhaps by adding a third section - Discussion (1 Introduction, 2 Effects of ENN and DON co-occurrence…, 3 Is missing at the moment? and 4 Conclusions). Otherwise I have no significant objections for article to be published.

Author Response

This interesting article deals with frequent Fusarium mycotoxins contaminating cereals all over the world - deoxynivalenol (DON), one of the major mycotoxins, and enniatins (ENNs), group of the emerging mycotoxins, providing an overview of the effects of their (simultaneous) exposure, emphasizing the need for more elaborate investigation of the interaction mechanisms of mycotoxin co-occurrence. The article seems to be well prepared, with the introduction providing enough background on the subject. The reference list is substantial, not entirely current but relevant, while English language and style are generally satisfying.

The authors are very grateful to reviewer 4 for the revision that has surely allowed us to greatly increase the scientific level of the manuscript.

The Authors could consider to shorten the Conclusions section for more clarity, perhaps by adding a third section - Discussion (1 Introduction, 2 Effects of ENN and DON co-occurrence…, 3 Is missing at the moment? and 4 Conclusions). Otherwise I have no significant objections for article to be published.

We revised accordingly. We added the section of Discussion in which we interpreted the main results and hypothesized future works. Moreover we shortened the Conclusions. Thanks for your suggestion.
